# Chronic Intermittent Hypoxia Increases Cell Proliferation in Hepatocellular Carcinoma

**DOI:** 10.3390/cells11132051

**Published:** 2022-06-28

**Authors:** Lydie Carreres, Marion Mercey-Ressejac, Keerthi Kurma, Julien Ghelfi, Carole Fournier, Olivier Manches, Florent Chuffart, Sophie Rousseaux, Mélanie Minoves, Thomas Decaens, Herve Lerat, Zuzana Macek Jilkova

**Affiliations:** 1Institute for Advanced Biosciences, Inserm U 1209, CNRS UMR 5309, Université Grenoble Alpes, 38000 Grenoble, France; lydie.carreres@univ-grenoble-alpes.fr (L.C.); mressejac@chu-grenoble.fr (M.M.-R.); keerthi.kurma143@gmail.com (K.K.); jghelfi@chu-grenoble.fr (J.G.); carole.fournier@inserm.fr (C.F.); olivier.manches@efs.sante.fr (O.M.); florent.chuffart@univ-grenoble-alpes.fr (F.C.); sophie.rousseaux@univ-grenoble-alpes.fr (S.R.); tdecaens@chu-grenoble.fr (T.D.); 2Service d’Hépato-Gastroentérologie, Pôle Digidune, CHU Grenoble Alpes, 38700 La Tronche, France; 3Service de Radiologie, Pôle Digidune, CHU Grenoble Alpes, 38700 La Tronche, France; 4Etablissement Français du Sang, Rhone-Alpes Auvergne, 38000 Grenoble, France; 5Laboratoire HP2, INSERM 1300, 38700 La Tronche, France; mminoves@chu-grenoble.fr; 6Secteur Essais Clinique, Pôle Pharmacie, CHU Grenoble Alpes, 38700 La Tronche, France; 7Unité Mixte de Service hTAG, Université Grenoble Alpes, Inserm US046, CNRS UAR2019, 38700 La Tronche, France

**Keywords:** obstructive sleep apnea syndrome, intermittent hypoxia, DEN-induced rat model, hepatocellular carcinoma, cell proliferation

## Abstract

Obstructive sleep apnea (OSA) syndrome is characterized by chronic intermittent hypoxia and is associated with an increased risk of all-cause mortality, including cancer mortality. Hepatocellular carcinoma (HCC) is the most common type of liver cancer, characterized by increasing incidence and high mortality. However, the link between HCC and OSA-related chronic intermittent hypoxia remains unclear. Herein, we used a diethylnitrosamine (DEN)-induced HCC model to investigate whether OSA-related chronic intermittent hypoxia has an impact on HCC progression. To elucidate the associated mechanisms, we first evaluated the hypoxia status in the DEN-induced HCC model. Next, to simulate OSA-related intermittent hypoxia, we exposed cirrhotic rats with HCC to intermittent hypoxia during six weeks. We performed histopathological, immunohistochemical, RT-qPCR, and RNA-seq analysis. Chronic DEN injections strongly promoted cell proliferation, fibrosis, disorganized vasculature, and hypoxia in liver tissue, which mimics the usual events observed during human HCC development. Intermittent hypoxia further increased cell proliferation in DEN-induced HCC, which may contribute to an increased risk of HCC progression. In conclusion, our observations suggest that chronic intermittent hypoxia may be a factor worsening the prognosis of HCC.

## 1. Introduction

Obstructive sleep apnea (OSA) syndrome is a sleep-related breathing disorder characterized by repeated episodes of partial or complete upper airway obstructions while a person is asleep. The main hallmark of OSA, responsible for major OSA-related comorbidities, is chronic intermittent hypoxia [1]. Chronic intermittent hypoxia is characterized by repetitive cycles of hypoxia and subsequent reoxygenation. Accordingly, the simulation of intermittent hypoxia in vitro and in vivo permits detailed analysis of the mechanisms responsible for the main deleterious effects of OSA.

Importantly, clinical and epidemiological reports suggest that OSA is independently associated with an increased risk of all-cause mortality, including cancer mortality [2,3]. In particular, studies focused on the impact of chronic intermittent hypoxia on cancer progression and clinical outcome have been gaining considerable attention [4,5]. The results from animal studies revealed that intermittent hypoxia promotes melanoma [6,7] and lung tumor progression [8] and increases the risk of a metastatic phenotype in breast cancer [9]. Accelerated tumor growth under intermittent hypoxia was also found to be associated with an increase in hypoxia-inducible factor-1 (HIF-1) [7] and VEGF [8,10].

Liver cancer is the third most common cause of cancer-related death worldwide [11], and its incidence is increasing. Hepatocellular carcinoma (HCC) represents the majority of primary liver cancers and is mainly caused by viral hepatitis, chronic alcohol consumption, non-alcoholic steatohepatitis (NASH), or aflatoxin intoxication. An emerging body of evidence indicates that OSA is associated with NASH and may contribute to metabolic-related HCC [12,13,14]. However, whether intermittent hypoxia can cause an acceleration in HCC progression, independent of NASH, remains unclear.

The diethylnitrosamine (DEN) chronically induced rat model reproduces the pathogenesis of HCC, as seen in humans. This includes liver damage, chronic inflammation, hepatocytes proliferation, liver fibrosis and cirrhosis, disorganized vasculature, and modulations of the liver’s immune microenvironment [15,16,17]. In general, the animal model of HCC mimics particularly well the “high proliferation” subtype of human HCC, without additional metabolic liver disease.

In this study, we investigated hypoxia pathway alterations in the DEN-induced HCC rat model during liver carcinogenesis and tested the impact of chronic intermittent hypoxia on already developed HCC. Our results provide support for a hypothesis of a link between OSA-related intermittent hypoxia and cancer progression. Particularly, our study revealed that chronic intermittent hypoxia may increase the risk of HCC progression and aggressiveness by accelerating tumor cell proliferation.

## 2. Materials and Methods

### 2.1. Animal Housing

Six-week-old Fischer 344 male rats were purchased from Charles River Laboratories (L’Arbresle, France) and housed at Plateforme de Haute Technologie Animale (PHTA), University of Grenoble-Alpes’ core facility (Grenoble, France). The stabilization time after the rats’ transportation was 1 week. The rats were kept in a temperature-controlled environment with a 12 h light/dark cycle and ad libitum access to water and diet under specific pathogen-free conditions. Animal housing and procedures were organized in accordance with the recommendations from the Direction des Services Vétérinaires, Ministry of Agriculture of France, the European Communities Council Directive 2010/63/EU, and the recommendations for health monitoring from the Federation of European Laboratory Animal Science Associations. The protocols were reviewed by the ethics committee “Comité d’Ethique pour l’Expérimentation Animale no.#12, Cometh-Grenoble” and approved by the French Ministry of Research, dated 04 January 2018 and 15 June 2020 (#12900-2018010416469902 v4 and #266054-2020061517289458 v2).

### 2.2. DEN-Induced Rat Model of HCC

Seven-week-old Fischer 344 male rats were treated once a week with intra-peritoneal injections of 50 mg/kg DEN (Sigma-Aldrich, Steinheim am Albuch, Germany) diluted in pure olive oil (Sigma-Aldrich, Steinheim am Albuch, Germany) in order to obtain HCC on a fibrotic/cirrhotic liver after 14 weeks, as published previously [15]. We analyzed three different time points of DEN treatment: (i) the 8-week group was sacrificed after 8 weeks of DEN injection, (ii) the 14-week group was sacrificed after 14 weeks of DEN injection, and (iii) the 14 + 6-week group was sacrificed after 14 weeks of DEN injection followed by 6 weeks without any DEN injections. Animals of the same age, without any DEN treatment, were used as controls (Figure 1a).

### 2.3. Intermittent Hypoxia Exposure Procedure

Seven-week-old Fischer 344 male rats were first treated weekly during 14 weeks with DEN in order to obtain HCC. The animals were then divided into two equivalent groups based on their visible nodule/tumor number and size, as assessed by ultrasound (VisualSonics Vevo^®^ 2100 Imaging System). During the following period of 6 weeks, one group of rats was exposed to intermittent hypoxia (IH), while the second group of rats was exposed to normoxia (N) in the HypE platform (HP2 laboratory, INSERM 1300, Grenoble-Alpes University). Briefly, in the IH group, the rats were exposed daily to 8 consecutive hours of 1 min IH cycles consisting of 30 s at 21% fractional inspired oxygen concentration and 30 s at 5% FiO_2_, as previously described [18]. The rats were exposed during their daytime sleeping period. In the control normoxia group, the rats were exposed to identical experimental chambers under normoxic air–air cycles. The fractional inspired oxygen level in the housing cages was monitored throughout the experiments (ML206 gas analyzer; AD Instruments, Oxford, UK). At the end of experiment, all rats were sacrificed and randomly fed using a ketamine–xylazine combination overdose. The blood was collected from the vena cava to measure the hematocrit. The serum was tested for liver safety markers (glucose, alkaline phosphatase (ALP), alanine transaminase (ALT), aspartate transaminase (AST), total bilirubin, albumin, cholesterol, and gamma glutamyltransferase (GGT)) (Table 1) by Charles River Clinical Pathology Services using Olympus instruments. The rats’ organs were weighed, visible nodules on the liver surface were counted. and the diameters of the five biggest nodules were measured. The liver triglycerides were measured as described previously [17].

### 2.4. Immunohistochemical and Immunofluorescence Analyses

The liver tissues were fixed in 10% formalin, neutral buffered (Sigma-Aldrich, Steinheim am Albuch, Germany), and paraffin-embedded, and four-micrometer sections of the tissue were prepared. The hematoxylin staining was used for histopathological examination (MHS32-1 L, Sigma-Aldrich, Steinheim am Albuch, Germany). To detect proliferating cells, the sections were incubated overnight at 4 °C with the primary anti-Cyclin D1 antibody (rabbit mAb, clone EPR2241, Abcam, diluted 1/200 in phosphate buffered saline (PBS) with 1% bovine serum albumin (BSA)). The anti-rabbit EnVision system horseradish peroxidase (HRP) Labelled Polymer (Dako Agilent, Santa Clara, CA, USA) was followed by 3,3′-Diaminobenzidine (DAB) for immune detection. The positive cells were counted on 10 randomly selected fields/sections captured by the Olympus BX41 microscope equipped with an Olympus DP70 camera. Collagen was detected with a picro-sirius red stain solution (Sigma-Aldrich, Steinheim am Albuch, Germany). Images were captured by the Olympus BX41 microscope equipped with an Olympus DP70 camera. To detect vascularization, paraffin-embedded liver sections were incubated with an anti-rat CD34 antibody (goat pAb, AF4117, R&D systems, diluted 1/100 in PBS with 1% BSA) followed by incubation with Alexa 647-conjugated donkey anti-goat IgG (Life Technologies, Carlsbad, CA, USA). Images were captured by an AxioCam MRm camera using the ApoTome microscope (Carl Zeiss, Oberkochen, Germany) and collected by AxioVision software. To detect hepatocarcinogenesis, paraffin-embedded liver sections were incubated overnight at 4 °C with the primary anti-GST-P (rabbit pAb, 311, MBL International, Woburn, MA, USA, diluted 1/2000 in PBS with 1% BSA) followed by the anti-rabbit EnVision system HRP Labelled Polymer (Dako Agilent, Santa Clara, CA, USA) and by DAB for immune detection. The positive area threshold was quantified using ImageJ software (NIH, Bethesda, MD, USA) on 10 to 15 randomly selected fields/sections (10× objective) and analyzed in a double-blinded manner.

### 2.5. Quantitative Real Time Polymerase Chain Reaction (RT-qPCR)

RNA was extracted from 30 mg of homogenized liver tissue with the RNesay Mini Kit^®^ (Qiagen, Hilden, Germany), following the manufacturer’s instructions. The RNA was quantified using the nanodrop, and the quality was accessed by the agilent 2100 bioanalyzer (Agilent, Waldbronn, Germany) to determine the RNA integrity number. Reverse transcription was performed using 1 μg of RNA, with the RIN above 9.5, with the SuperScript™ IV VILO™ Master Mix with the ezDNase enzyme (Thermo Fisher), according to the manufacturer’s instructions. The qPCR was performed using the predesigned TaqMan probe (Thermo Fisher, Rockford, IL, USA, Appendix A) to assess the target genes involved in angiogenesis, extracellular matrix remodeling, and cell proliferation using a Thermocycler sequence detector (BioRad CFX96, Hercules, CA, USA). The target genes expression levels were normalized to Hprt1 expression levels and quantified using the 2-ΔΔCT method.

### 2.6. RNA-Seq

The total RNA was extracted from the rat liver tissue samples preserved with an RNA stabilization solution (Thermo scientific, USA). RNA purification was performed with the RNeasy Mini Kit^®^ (Qiagen, CA, USA), according to the instructions provided by the manufacturer. Quality assessment for the RNA Integrity Number (RIN) was performed using the Agilent 2100 Bioanalyzer (Agilent, Palo Alto, CA, USA). The stranded mRNA library was prepared using 1 μg total RNA (RIN > 7). The normalized and pooled libraries were sequenced on the DNBseqTM platerform (MGI). The fastq files were aligned on the UCSC rn6 genome using the STAR (2.7.1a) to produce bam files [19], which were counted using the HTSeq framework (0.11.2) [20] (with options: -t exon -f bam -r pos –stranded = reverse -m intersectiostrict –nonunique none). Normalization and differential analysis were performed using the R software (R Core Team. R: A language and environment for statistical computing; Vienna, Austria: R Foundation for Statistical Computing, 2017) and the SARTools [21] and DESeq2 (1.22.2) [22,23] packages. The raw sequencing data from the DEN-induced rat model as well as for the intermittent hypoxia versus the Normoxia-treated rats are available at NCBI’s Gene Expression Omnibus under the GEO Series accession number GSE182860. GSEA was conducted with the pre-ranked GSEA method within the Hallmark databases (https://broadinstitute.org/msigdb, accessed on 10 June 2022). Single-sample GSEA scores, representing the degree to which the genes in a gene set are coordinately up- or down-regulated within a sample, were calculated using the GSVA program [24], following the methodology described previously [25].

### 2.7. Statistical Analysis

All of the data were tested for normality (D’Agostino–Pearson normality test). Comparisons of the means were calculated by using ANOVA tests with the Tukey HSD correction for multiple means comparisons. An independent *t*-test was used only when two means were compared. Statistical analyses were performed using Prism 9 (GraphPad Software Inc., San Diego, CA, USA). The data are presented as the mean values ± standard error mean (SEM).

## 3. Results

### 3.1. Chronic DEN Promotes Hypoxia in Liver

Fisher 344 rats were injected with DEN each week for 14 weeks. This induced progressive liver damage and liver carcinogenesis, leading to nodule development in 100% of the animals after 14 weeks and to fully developed HCC at 14 + 6 weeks, as illustrated in Figure 1b. Confirming previous results [15], a DEN treatment strongly promoted hepatocyte proliferation, as assessed by Cyclin D1 staining (Figure 1c, left panels). Similarly, chronic DEN injections led to the development of fibrosis/cirrhosis, as revealed by Sirius red staining (Figure 1c, middle panels). This was associated with disorganized, abnormal vascular patterns with a significant expansion of the CD34 positive area in the DEN-treated rat liver (Figure 1c, right panels) compared to the animals of the same age without any DEN treatment (NoDEN rats).

Since liver fibrosis and abnormal vasculature, which are usually observed during HCC development, are often also associated with tissue hypoxia [26,27], we decided to use RNA sequencing (RNA-seq) in order to investigate in more detail how a chronic DEN treatment modulates the hypoxia status in liver tissue and in HCC.

The single-sample gene set enrichment analysis (GSEA) of our rat model using a Hypoxia hallmarks gene set revealed the up-regulation of genes involved in the response to low oxygen levels following a DEN treatment (Figure 1d). Comparing the non-tumoral liver tissue between the DEN-treated and age-matched untreated (NoDEN) rats at 14 weeks (Figure 1e) and 14 + 6 weeks (Figure 1f), the GSEA plots show an enrichment of the Hypoxia hallmarks gene set. Importantly, a GSEA plot of the same gene set comparing the tumor and non-tumor tissues of the DEN-treated rats at 14 + 6 weeks also shows an enrichment (Figure 1g). The subsets of HIF target genes that contribute most to these enrichment results are presented in Appendix A and S3.

Altogether, these results show that DEN-induced liver damage and carcinogenesis are associated with hypoxia in non-tumoral liver tissue as well as in HCC, which is a typical feature of human HCC associated with increased aggressiveness.

### 3.2. Intermittent Hypoxia Increases Cell Proliferation in Chronic DEN-Induced HCC

Next, we investigated whether obstructive sleep apnea (OSA)-related chronic intermittent hypoxia could further influence HCC progression and aggressiveness. For this purpose, the rats were treated with DEN weekly for 14 weeks in order to develop HCC. The nodule/tumor number and size were evaluated by ultrasound, and the animals were then divided into two equivalent groups (*n* = 9/group). During the following period of 6 weeks, the rats were exposed to intermittent hypoxia (IH group) or to normoxia (N group) during eight hours of their sleeping period (Figure 2a).

The effect of chronic intermittent hypoxia was first evaluated on the hematocrit levels. The hematocrit, corresponding to the volume of red blood cells over the volume of total blood, was significantly higher in the intermittent hypoxia group compared to normoxia (48 ± 1.9 versus 40 ± 1.5, *p* = 0.0040), which is a well-known physiological response to chronic hypoxia [28,29]. The total body weight, just as the weight of the liver, heart, spleen, or visceral adipose tissue, did not vary significantly between the intermittent hypoxia or normoxia groups (Table 1). The blood sample analysis revealed that intermittent hypoxia significantly decreased the glucose level compared to that in the normoxia-treated rats. No difference between the groups was observed in alkaline phosphatase (ALP), alanine transaminase (ALT), aspartate transaminase (AST), total bilirubin, or albumin levels. The exposure to intermittent hypoxia did not affect the cholesterol or triglyceride blood concentrations nor the gamma-glutamyltransferase (GGT) level (Table 1).

Next, we evaluated the effect of intermittent hypoxia on HCC progression. We observed no difference in the incidence and size of the tumors between the IH group and the N group at 14 + 6 weeks (Figure 2c). Nevertheless, the GST-P+ lesions were larger in the IH group compared to those in the N group (34.04 ± 2.15% versus 26.43 ± 1.81%, *p* = 0.0176) (Figure 2d). Consistently, intermittent hypoxia treatment significantly promoted tumor cell proliferation, as assessed by Cyclin D1 staining (Figure 2e). More precisely, the number of Cyclin D1 positive cells in the tumoral area was higher in the IH group compared to the N group (32.43 ± 2.93% versus 22.47 ± 2.9%, *p* = 0.0286); a trend was observed in the non-tumor area (2.85 ± 0.58% versus 1.42 ± 0.38%, *p* = 0.0562). This was consistent with the increased expression of the Ccnd1 gene encoding the Cyclin D1 protein in the IH group compared to the N group (Figure 2e). The promoting effect of intermittent hypoxia on cell proliferation and cell cycle progression in the tumor tissue was further confirmed by PCR analyses, revealing that genes considered as markers of these processes, including genes encoding the PDZ binding kinase (Pbk) and MYB proto-oncogene like 2 (Mybl2), are highly upregulated in the IH tumor samples (Figure 2g).

CD34 staining did not reveal any significant alteration of the vascular pattern associated with the intermittent hypoxia treatment (Appendix A). Similarly, we did not observe any difference in the severity of the fibrosis/cirrhosis assessed by Sirius red staining (Appendix A). The expression of the Collagen-1 and alpha smooth muscle actin (α-SMA) genes, as assessed by RT-qPCR, did not significantly increase in the IH group (Appendix A), nor did the expression of the genes encoding the transforming growth factor beta (TGF-β), Collagen-5, or the matrix metalloproteinase 2 (Mmp2). The expression profiles associated with intermittent hypoxia in the liver were further investigated by RNA sequencing (RNA-seq), comparing the non-tumor liver tissue between the intermittent hypoxia group (IH-NT) and the normoxia group (N-NT). GSEA revealed that the gene sets significantly enriched (upregulated genes) in liver following the intermittent treatment are those containing genes involved in the cell cycle division and proliferation, with the three top gene sets being the G2M checkpoint, the E2F targets, and the mitotic spindle assembly (Figure 3).

This observation is in line with the immunohistochemistry results, demonstrating that chronic intermittent hypoxia induces cell proliferation. A similar GSEA comparing the tumor tissue between rats exposed to intermittent hypoxia (IH-T) and rats under normoxia (N-T) also identified a significant enrichment in the gene sets related to the G2M checkpoint and the E2F targets. It also revealed a significant enrichment in genes regulated by MYC, which is one of the oncogenes known to promote tumor cell proliferation [30]. Indeed, both the MYC targets v1 and MYC targets v2 gene sets were enriched in tumors from intermittent hypoxia-treated rats (Figure 3b). The subsets of genes that contribute the most to these enrichment results are presented in Appendix A. Additionally, GSEA identified other significantly enriched gene sets, including those related to oxidative phosphorylation and the activation of mTORC1 complex signaling. Altogether, these results suggest that intermittent hypoxia increases tumor cell proliferation, which can accelerate HCC aggressiveness.

Based on the GSEA, no significant modulation in the Hypoxia hallmarks gene set was observed in the liver tissue from the intermittent hypoxia group compared to the normoxia group. Similarly, the PCR analysis did not reveal any significant differences in HIF signaling between the two groups. For instance, we observed only a trend toward the upregulated expression of erythropoietin (EPO) in the IH liver compared to the N liver (1.63 ± 0.61 versus 1.00 ± 0.19, *p* = 0.3609) (Appendix A). Considering the fact that liver tumor tissue from the DEN-induced model is itself characterized by a high hypoxia status (Figure 1), the possibility of observing a more pronounced activation of the hypoxia pathway in response to intermittent hypoxia stimulation is limited, although the increased hematocrite levels clearly demonstrated a systemic physiological response of the rats to intermittent hypoxia exposure. Moreover, the liver tissues were collected 12 h after the last hypoxia cycle, which could affect the transient effects of intermittent hypoxia.

## 4. Discussion

OSA is associated with obesity, which worsens metabolic liver diseases [14,31,32]. The link between OSA-related intermittent hypoxia and metabolic-related HCC is currently being actively investigated. However, whether chronic intermittent hypoxia can accelerate HCC progression, independent of NASH, remains unclear. In this study, we examined the effects of chronic intermittent hypoxia on the tumor progression in the absence of obesity.

The present animal study, performed under controlled conditions to investigate the effect of chronic intermittent hypoxia, provides support for a hypothesis of a link between intermittent hypoxia, a hallmark of OSA, and cancer progression. Solid tumors are generally characterized by hypoxia. With regard to HCC, the most aggressive tumor subtype is the proliferative class that accounts for ~50% of HCC and shows the worst prognosis [33]. The DEN-induced cirrhotic rat model of HCC particularly recalls the proliferative subclass [15]. In the present study, we demonstrate that the hypoxia signaling pathway is involved in this model during the progression to HCC. Six weeks of intermittent hypoxia stimulus did not affect the tumor size. However, our study revealed that OSA-related chronic intermittent hypoxia accelerates tumor cell proliferation, possibly by involving MYC pathways. This may increase the risk of HCC progression and aggressiveness in the long term, but the exact mechanism remains to be clarified. MYC-driven HCC is usually associated with a poor prognosis [34], and it would be therefore important to verify the possible dysregulation of the MYC oncogene in HCC patients with OSA in the future. The results of the present study also suggest that chronic intermittent hypoxia leads to modulation in the glucose metabolism in the DEN-induced rat model. This is in accordance with previously reported studies showing chronic intermittent hypoxia–related alterations in fasting glucose levels and glucose metabolism in general [35,36,37].

One of the limitations of this study is the fact that the tissues and blood samples were collected 12 h after the last hypoxia cycle. Therefore, we could not precisely evaluate the direct transient effects of intermittent hypoxia. For instance, intermittent hypoxia is known to stabilize HIF-1α, which results in a rapid increase in EPO [38]. Similarly, recent studies revealed that recurrent cycles of hypoxia lead to an upregulation of HIF2α [39] and that HIP/PAP is modulated in response to intermittent hypoxia [40]. Our results did not reveal any major modulations of HIF signaling in liver tissue, which is probably related to the type of experimental model and the selected protocol. Herein, we investigated only the effect of chronic intermittent hypoxia on the progression of HCC, but its potential effect on the HCC development should be addressed in future research. Our results suggest that a longer exposure to intermittent hypoxia stimulus, already starting during the chronic DEN injections, i.e., before HCC development, may increase both the number and size of tumors. Additionally, the future research effort should also focus on the effect of chronic intermittent hypoxia on HCC metastasis development using an appropriate HCC animal model.

The model of chronic hypoxia used in this study did not involve hypercapnia, a common feature observed in obstructive sleep apnea patients, and the hypoxia conditions used in the study protocol are rather severe, which are factors that must be taken into account when this model is compared with obstructive sleep apnea patients.

In conclusion, our study reveals that the exposure to chronic intermittent hypoxia enhances tumor cell proliferation, which may contribute to HCC aggressiveness. Thus, intermittent hypoxia could be considered as a factor worsening the prognosis of HCC.

## Figures and Tables

**Figure 1 cells-11-02051-f001:**
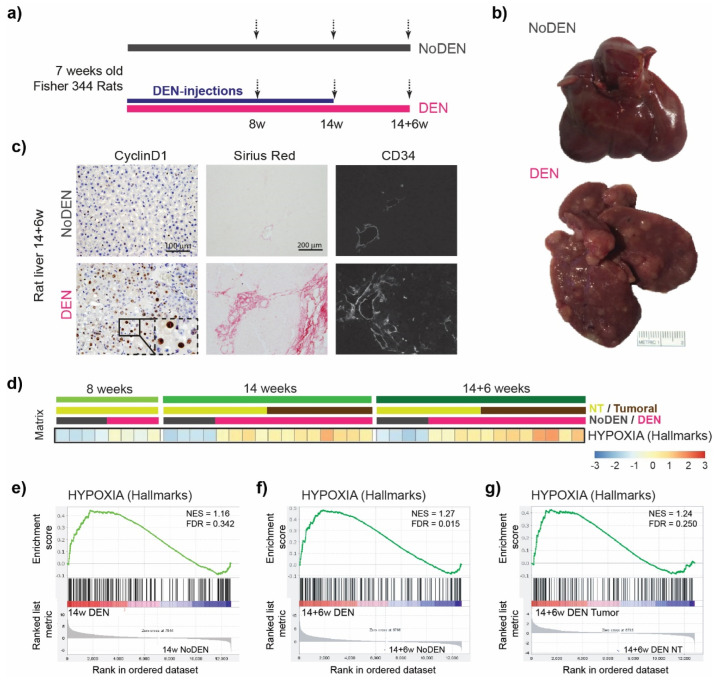
Chronic DEN-induced hepatocarcinogenesis is associated with hypoxia. (**a**) Timeline protocol of the DEN-induced cirrhotic rat model of HCC. Seven-week-old Fisher 344 rats were injected weekly by DEN (50 mg/kg per week); 8 w indicates 8 weeks of DEN injections, 14 w indicates 14 weeks of DEN injections, and 14 + 6 w indicates 14 weeks of DEN injections followed by 6 weeks of no DEN injections. The arrows indicate the day of tissue harvest. (**b**) Representative pictures of the liver at 14 + 6 w from the rat without DEN treatment (upper panel) and the rat chronically treated with DEN (lower panel). (**c**) Representative images of the liver from rats untreated (upper panels) or treated with DEN injections (as above, lower panels). Histopathological sections stained with nuclear Cyclin D1 (20× magnification, left panels), with Sirius red (middle panels), and with anti-CD34 immunofluorescence (right panels). (**d**) Heatmap showing the single-sample Gene Set Enrichment Analysis (GSEA) scores for the Hypoxia Hallmarks gene set. (**e**–**g**) GSEA plots of differentially expressed genes from livers of DEN-treated versus untreated (NoDEN) rats at 14 weeks ((**e**): non-tumor tissue) and 14 + 6 weeks ((**f**): non-tumor tissue, (**g**): tumor tissue), showing the enrichment of genes associated with the response to low oxygen levels (Hypoxia-Hallmarks). NES, normalized enrichment score; FDR, false discovery rate; w, weeks.

**Figure 2 cells-11-02051-f002:**
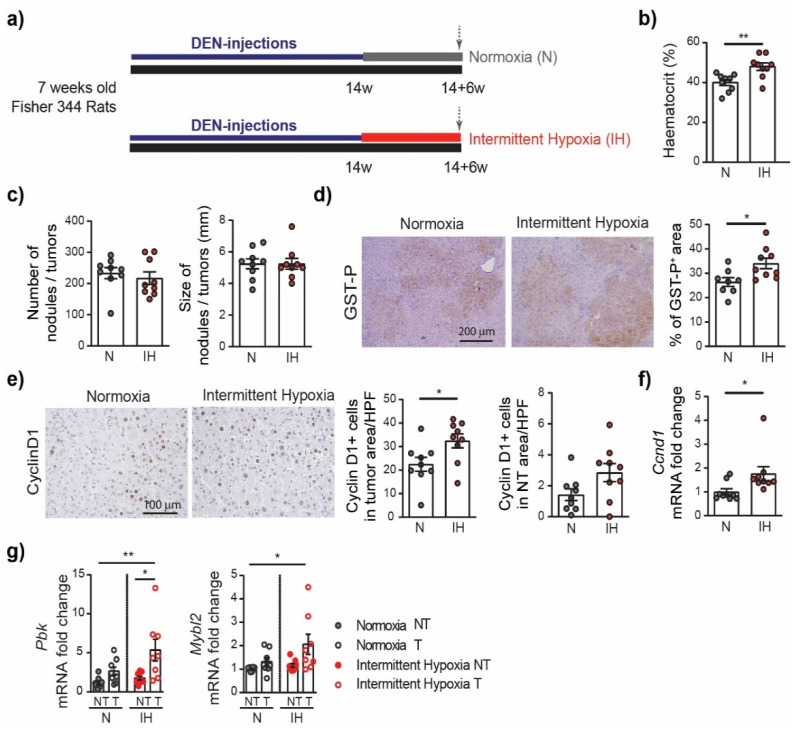
Intermittent hypoxia increases cell proliferation in chronic DEN-induced HCC. (**a**) Timeline protocol of the intermittent hypoxia experiment. Seven-week-old Fisher 344 rats were injected weekly by DEN (50 mg/kg per week) during 14 weeks and then subjected to either normoxia or intermittent hypoxia during 6 weeks. The arrows indicate the day of tissue harvest. (**b**) Effect of chronic intermittent hypoxia on the haematocrit (the volume percentage (%) of red blood cells in the blood of rats). (**c**) Macroscopic examination of livers with the assessment of the tumor number at the surface of livers and the tumor size (average of the diameter of the five largest tumors). (**d**) Representative images of GST-P staining (10× magnification) and quantification of GST-P+ surface area per high power field (HPF). (**e**) Representative images of nuclear Cyclin D1 staining (20× magnification) and quantification of nuclear Cyclin D1 positive hepatocytes per tumor area or per non tumoral area per HPF. (**f**) qPCR analysis of Cyclin D1 gene Ccnd1 expression in the liver samples. Each circle represents an individual animal, mean ± SE, *n* = 9/group. The comparison of the means was performed by an unpaired *t*-test. (**g**) The impact of intermittent hypoxia on cell proliferation and cell cycle progression, as determined by the gene expression of PDZ binding kinase (Pbk) and MYB proto-oncogene like 2 (Mybl2). NT—non-tumor tissue, T—tumoral tissue, *n* = 8–9/group. The comparison of the means was performed by ANOVA with the Tukey HSD correction for multiple means comparisons, * *p* < 0.05, ** *p* < 0.01.

**Figure 3 cells-11-02051-f003:**
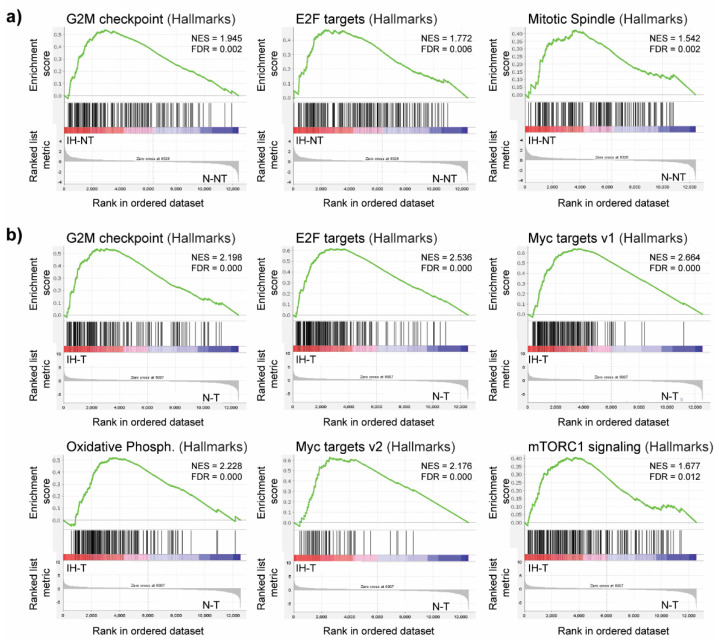
GSEA plots showing upregulated gene sets in the livers of DEN-treated rats following intermittent hypoxia treatment. (**a**) Enriched gene sets induced by intermittent hypoxia in non-tumor liver tissue (IH-NT) compared to normoxia (N-NT) in non-tumor liver tissue. (**b**) Enriched gene sets induced by intermittent hypoxia in tumor tissue (IH-T) compared to normoxia (N-T) in tumor tissue. NES, normalized enrichment score; FDR, false discovery rate.

**Table 1 cells-11-02051-t001:** Clinical and biological analyses.

	Normoxia-Treated Rats, *n* = 9	Intermittent Hypoxia-Treated Rats, *n* = 9	*p* Value
Body weight (BW), g	242 [182–283]	242 [222–270]	0.560
Liver weight, g	13.5 [7.0–15.1]	13.9 [10.8–15.4]	0.743
Liver weight/BW, %	5.26 ± 0.29	5.39 ± 0.29	0.782
Heart weight, mg	754.8 ± 33.5	741.6 ± 35.2	0.789
Spleen weight, mg	1081 ± 67.9	987.3 ± 28.8	0.224
Visceral adipose tissue, g	1.68 ± 0.29	1.58 ± 0.08	0.754
Glucose, mmol/L (before sacrifice)	5.1 ± 0.1	4.4 ± 0.1	<0.0001
ALP, IU/L	200.9 ± 7.2	213.3 ± 8.7	0.286
ALT, IU/L	156.8 ± 17.3	196.0 ± 17.5	0.131
AST, IU/L	329.6 ± 62.2	365.6 ± 56.7	0.675
Total Bilirubin, µmol/L	3.20 (0.1–14.4)	3.4 [1.2–5.5]	>0.999
Albumin, g/L	37.2 ± 0.7	37.6 ± 0.4	0.605
Cholesterol, mmol/L	3.4 ± 0.2	3.2 ± 0.2	0.509
Triglycerides, mmol/L	0.63 ± 0.06	0.91 ± 0.14	0.096
GGT, IU/L	50.7 ± 4.8	55.8 ± 5.4	0.491
Liver Triglycerides, mmol/L	20.3 ± 1.5	21.3 ± 1.0	0.567

Alkaline Phosphatase, ALP; Alanine transaminase, ALT; Aspartate transaminase, AST; Gamma Glutamyltransferase, GGT.

## Data Availability

The raw sequencing data from the DEN-induced rat model are available at NCBI’s Gene Expression Omnibus under the GEO Series accession number GSE182860 (https://www.ncbi.nlm.nih.gov/geo/query/acc.cgi?acc=GSE182860, accessed on 31 December 2020).

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
