# Peer review of "Chronic Intermittent Hypoxia Increases Cell Proliferation in Hepatocellular Carcinoma"

_cells, 2022, doi:10.3390/cells11132051_

Round 1
Reviewer 1 Report
The manuscript of Carreres et al. investigated the impact of chronic intermittent hypoxia on a hepatocellular carcinoma rat model. The topic is very interesting and the model is well chosen, but some things need to be presented and discussed in more detail. In particular, the influence of hypoxia must be considered in more detail. Below are a few selected examples.
Abstract:
Line 17-21: The relationship between OSA and HCC is not clear here. This needs to be worked out more clearly.
Line 21: Indroduce DEN
What’s the rationale behind investigating hypoxia in the HCC model?
Introduction:
Line 49: What´s with HIF2a? Chronic hypoxia and recurrent cycles of hypoxia led to an upregulation of predominantly HIF2a (see for example: PMID: 35159368). You should address this not only in the introduction.
Line 65: IH must be introduced. Have you seen what happens when you apply IH and DEN at the same time? That would also be very exciting to see if it contributes to tumor formation. Is anything known about this?
Materials and Methods:
Line 85: At what age was the treatment started?
Line 90: The designation of group iii) is unfavorable. Wouldn't it be better to call it 14 weeks plus 6 week observation, or something like that?
Line 108-110: Methodology not entirely clear. Abbreviations need to be introduced.
Line:115-136: For some antibodies, the article number is missing and the dilution used should be added.
Results:
Line 190: NoDEN rats: Must be introduced
Line 191-194: This section is unclear. Which event is meant and how exactly is it related to hypoxia. Please explain more clearly. This is important so that it is clear why.
Line 195-204: Which hypoxia hallmarkers were regulated exactly? HIF1a or Hif2a? Hif target gene? Can you add this to the figure and address it accordingly.
Figure 2: Have you looked into EPO? EPO is a HIF target gene. Cyclin D1 is a HIf target gene. I would explain this in the text.
Line 252-253: Was that expected? HIF regulates GLUT1. You need to discuss that in the discussion section
Line 274-275: Have you checked for Ki67? How have you selected these markers?
Figure 3: This generally refers to a methodological aspect. How long after the last hypoxia cycle were the tissues removed? This has not become entirely clear and is important for interpreting the results here. Have no hypoxia genes been regulated at all? Can you look at this in more detail? See also comment above.
Line 305: Have you looked for metastases in the rats? Instead of aggressiveness, which somehow also implies metastasis, I would rather write progression.
Discussion:
The discussion needs to be completely revised. At the moment, it is only a summary of the results, but not a critical analysis of them. What are the limitations of the study, how can the data be interpreted?
Line 316: What role does obesity play in this context now?
Author Response
Reviewer 1
Comments and Suggestions for Authors
The manuscript of Carreres et al. investigated the impact of chronic intermittent hypoxia on a hepatocellular carcinoma rat model. The topic is very interesting and the model is well chosen, but some things need to be presented and discussed in more detail. In particular, the influence of hypoxia must be considered in more detail. Below are a few selected examples.
Response: We thank Reviewer #1 for the positive feedback, her/his insights and very constructive comments.
Abstract:
Line 17-21: The relationship between OSA and HCC is not clear here. This needs to be worked out more clearly.
Response: We thank Referee #1 for this comment. The sentence has been reformulated.
Line 21: Indroduce DEN
Response: We thank Referee #1 for this comment. DEN is now introduced in the abstract.
What’s the rationale behind investigating hypoxia in the HCC model?
Response: We wanted to understand if chronic intermittent hypoxia may be a one of the protective or rather detrimental factors for HCC progression. To investigate this question, we used the animal model that mimics human HCC and we needed first to understand how is the hypoxia status modified by DEN injections in this model – before or without the intermittent hypoxia challenge.
Introduction:
Line 49: What´s with HIF2a? Chronic hypoxia and recurrent cycles of hypoxia led to an upregulation of predominantly HIF2a (see for example: PMID: 35159368). You should address this not only in the introduction.
Response: We thank Referee #1 for the valuable suggestion. We have added this information in the revised manuscript. Discussion “recent studies revealed that recurrent cycles of hypoxia led to an upregulation of pre-dominantly HIF2α [Ref], and that..”
Line 65: IH must be introduced. Have you seen what happens when you apply IH and DEN at the same time? That would also be very exciting to see if it contributes to tumor formation. Is anything known about this?
Response: We apologize for the unclear description. “Intermittent hypoxia” is now used in whole text and IH is used only as a description of group treated by intermittent hypoxia.
Simultaneous treatment of intermittent hypoxia and DEN was not tested but this valuable suggestion is now included in the discussion.
Materials and Methods:
Line 85: At what age was the treatment started?
Response: We apologize for the unclear description of the age. The information is now included. “7-week-old Fischer 344 male rats..”
Line 90: The designation of group iii) is unfavorable. Wouldn't it be better to call it 14 weeks plus 6 week observation, or something like that?
Response: We thank Referee for the suggestion. The group is now named 14+6 as recommended. We agree that this modification increase the clarity of treatment protocols in whole manuscript, including modified Figure 1 and Figure 2.
Line 108-110: Methodology not entirely clear. Abbreviations need to be introduced.
Response: We thank Referee for this comment. We modified the text introducing all abbreviations as following: “Serum was tested for liver safety markers (glucose, alkaline phosphatase (ALP), alanine transaminase (ALT), aspartate transaminase (AST), total bilirubin, albumin, cholesterol and gamma glutamyltransferase (GGT)), Table 1, by Charles River Clinical pathology Services using Olympus instruments.”
Line:115-136: For some antibodies, the article number is missing and the dilution used should be added.
Response: We apologize for the uncompleted description of the antibodies. The article number and the dilution is added for each primary antibody.
Results:
Line 190: NoDEN rats: Must be introduced
Response: We apologize. The information is now corrected. “…compared to animals of the same age, without any DEN treatment (NoDEN rats).”
Line 191-194: This section is unclear. Which event is meant and how exactly is it related to hypoxia. Please explain more clearly. This is important so that it is clear why.
Response: We thank Referee #1 for this comment. The sentence is now clarified. “Since liver fibrosis and abnormal vasculature S, which are usually observed during cancer HCC development, are often also associated with tissue hypoxia [26,27], we decided to use RNA sequencing (RNA-seq) in order to investigate in more details how a chronic DEN treatment modulates hypoxia status in liver tissue and in HCC.”
Line 195-204: Which hypoxia hallmarkers were regulated exactly? HIF1a or Hif2a? Hif target gene? Can you add this to the figure and address it accordingly.
Response: We thank Referee #1 for this comment that encouraged us to investigate deeply the Hypoxia enriched gene sets. Indeed, we observed that HIF target genes are the predominantly regulated genes. The lists of hypoxia-related genes upregulated by DEN injections are now part of the supplementary results:
The main text “The subsets of HIF target genes that contribute most to these enrichment results are presented in Table S2-3. “ Supplementary results : Table S2: The subset of genes that contributes most to the enrichment result of Hypoxia gene set in 14+6w DEN liver versus NoDEN liver. Table S3: The subset of genes that contributes most to the enrichment result of Hypoxia gene set (14+6w DEN Tumor versus 14+6w DEN Non Tumoral)
Figure 2: Have you looked into EPO? EPO is a HIF target gene. Cyclin D1 is a HIf target gene. I would explain this in the text.
Response: We thank Referee #1 for this comment. EPO was measured by PCR on whole liver level and included in Supplementary Figure S1d. In the revised version, we comment this result in details in the text. Results part: “For instance, we observed only a trend toward upregulated expression of EPO in the IH liver compared to N liver (1.63 ± 0.61 versus 1.00 ± 0.19, p = 0.3609), Figure S1d.”
Additionally, we verified EPO in RNAseq data of liver and we observed same unsignificant modulation. This is probably related to the experimental protocol.
Reviewer 2 Report
· This is valid models for HCC rats sleep apnea
· It must be mentioned that the model of sleep apnea used has a major limitation since no physical obstruction is performed. Thus, hypercapnia, a major feature in obstructive sleep apnea patients, is lacking in this model and must be mentioned as a limitation. Moreover, a 5% hypoxia is considered severe form of sleep apnea that is not frequently seen in clinics of obstructive sleep apnea. The later should be mentioned as limitation as well.
· The animal model of sleep apnea used is well known to enhance systemic and local inflammation and oxidative stress (please see PMID: 30880336 DOI: 10.1093/sleep/zsz066 and doi: 10.1155/2019/4093018). Whether both factors could worsen the progression of HCC must be discussed. In other terms, is the worsening seen in HCC patients with sleep apnea specific to sleep apnea? Or any aggravator of systemic oxidative stress and inflammation could possess similar effects.
· Line 57-59: suggest rephrasing of the sentence
· Line 85-92: suggest rephrasing of the sentence and clarifying the methods, more specifically thos sentence “20-week group was sacrificed after 14 weeks of DEN injection followed by 6 weeks of no DEN injection”
Author Response
Reviewer 2
- This is valid models for HCC rats sleep apnea
Response: We thank Reviewer #2 for the positive feedback and her/his comments.
- It must be mentioned that the model of sleep apnea used has a major limitation since no physical obstruction is performed. Thus, hypercapnia, a major feature in obstructive sleep apnea patients, is lacking in this model and must be mentioned as a limitation. Moreover, a 5% hypoxia is considered severe form of sleep apnea that is not frequently seen in clinics of obstructive sleep apnea. The later should be mentioned as limitation as well.
Response: We thank Referee #2 for this criticism. The discussion is now modulated and the limits of this model are presented.
- The animal model of sleep apnea used is well known to enhance systemic and local inflammation and oxidative stress (please see PMID: 30880336 DOI: 10.1093/sleep/zsz066 and doi: 10.1155/2019/4093018). Whether both factors could worsen the progression of HCC must be discussed. In other terms, is the worsening seen in HCC patients with sleep apnea specific to sleep apnea? Or any aggravator of systemic oxidative stress and inflammation could possess similar effects.
Response: We thank Referee 2 for this suggestion and questions. DEN-induced rat model is by itself characterized by inflammation and oxidative stress which contributes to HCC development. We reanalyzed our RNAseq data and we did not observe any enrichment of the gene sets related to the inflammation and oxidative stress when the liver from the chronic intermittent hypoxia group was compared to normoxia group, suggesting that there is no IH-related increase of inflammation and oxidative stress in the liver that could be responsible for the enhanced proliferation of tumour cells. However, concerning systemic oxidative stress analysis or systemic inflammation analysis, we never studied the endothelial functions, endothelial nitric oxide uncoupling or circulating cytokine levels. Indeed, it could be a very interesting subject for the following study.
- Line 57-59: suggest rephrasing of the sentence
Response: This paragraph was rephrased.
- Line 85-92: suggest rephrasing of the sentence and clarifying the methods, more specifically thos sentence “20-week group was sacrificed after 14 weeks of DEN injection followed by 6 weeks of no DEN injection”
Response: We thank Referee 2 for this comment. The sentence is modified and the group is now named 14+6. We hope that this modification increase the clarity of treatment protocols in whole manuscript.
Reviewer 3 Report
Although the findings that chronic intermittent hypoxia (IH) increased cell proliferation in diethylnitrosamibe (DEN)-induced rat hepatocarcinomas are interesting, numbers of points need clarifying and certain statements require further justification. These are given below.
<Points>
1 The authors found chronic IH increased cell proliferation in rat model. However, precise mechanism how IH increased hepatocarcinoma proliferation was still unclear in this study. Additional experiments and/or discussion should be required.
2 Concerning hepatocellular carcinomas, Brechot, C. and his collaborators found HIP gene and its roles for hepatocellular carcinoma proliferation (Cancer Res. 52, 5089-5095, 1992; Eur. J. Biochem. 224, 29-38, 1994; Am. J. Pathol. 155, 1525-1533; Hepatology 53, 618-627, 2011). Iovanna, J.L.’ group found PAP gene(s) (J. Clin. Invest. 90,2284-2291, 1992; Biochim. Biophys. Acta 1216, 329-331, 1993; Biochemical J. 307, 9-16, 1995; Br. J. Cancer 74,1767-1775, 1996) and PAP and HIP were found to be the same gene (HIP/PAP) (Okamoto, H. & Takasawa, S. Proc. Jpn. Acad. Ser. B Phys. Biol. Sci. 97, 423-461, 2021). Was the gene expression of HIP/PAP changed in response to IH?
3 Uchiyama, T. et al. reported (Biochem. Biophys. Rep. 11, 130-137, 2017) that IH induced HIP/PAP expression in human hepatoma cells (HepG2, JHH5, and JHH7 cells) in cell culture system. In animal experiment, how the gene expression of Hip/Pap was affected by IH should be included.
4 In Citation (lines 28-30), Autor(s)’ name and title of manuscript should be added.
5 “IH” should be explained: intermittent hypoxia (IH). (line 65)
6 Although the authors showed approval number, they did not show approval date (lines 79-82). Please add the approval date.
7 “ImageJ software (NIH, USA)” (line 135) should be changed to “ImageJ software (NIH, Bethesda, MD)”.
8 “RNAeasy” (line 139) should be changed to “RNeasy”.
9 “Retrotranscription” (line 142) should be changed to “Reversetranscription”.
10 “Thermofisher” (lines 144, 145) should be changed to “Thermo Fisher”.
11 “GraphPad Software Inc., CA, USA” should be changed to “GraphPad Software Inc., San Diego, CA)”.
12 There was no DOI in Ref. 15.
Author Response
Reviewer 3
Comments and Suggestions for Authors
Although the findings that chronic intermittent hypoxia (IH) increased cell proliferation in diethylnitrosamibe (DEN)-induced rat hepatocarcinomas are interesting, numbers of points need clarifying and certain statements require further justification. These are given below.
Response: We thank Reviewer #3 for finding our study interesting and mainly for her/his insights and very constructive comments and suggestions.
1 The authors found chronic IH increased cell proliferation in rat model. However, precise mechanism how IH increased hepatocarcinoma proliferation was still unclear in this study. Additional experiments and/or discussion should be required.
Response: We thank Referee #3 for this suggestion. We included additional supplementary data that can help to generate further hypothesis related to the mechanism. “Table S4: The subset of genes that contribute to the enrichment result of Hallmark Myc Targets genesets induced by intermittent hypoxia in tumor tissue compared to tumor tissue under normoxia conditions.”
However, we agree with the reviewer that at this stage, the precise mechanism of intermittent hypoxia-enhanced tumor cell proliferation is not clarified. This point is now added in the discussion of the revised manuscript.
2 Concerning hepatocellular carcinomas, Brechot, C. and his collaborators found HIP gene and its roles for hepatocellular carcinoma proliferation (Cancer Res. 52, 5089-5095, 1992; Eur. J. Biochem. 224, 29-38, 1994; Am. J. Pathol. 155, 1525-1533; Hepatology 53, 618-627, 2011). Iovanna, J.L.’ group found PAP gene(s) (J. Clin. Invest. 90,2284-2291, 1992; Biochim. Biophys. Acta 1216, 329-331, 1993; Biochemical J. 307, 9-16, 1995; Br. J. Cancer 74,1767-1775, 1996) and PAP and HIP were found to be the same gene (HIP/PAP) (Okamoto, H. & Takasawa, S. Proc. Jpn. Acad. Ser. B Phys. Biol. Sci. 97, 423-461, 2021). Was the gene expression of HIP/PAP changed in response to IH? Uchiyama, T. et al. reported (Biochem. Biophys. Rep. 11, 130-137, 2017) that IH induced HIP/PAP expression in human hepatoma cells (HepG2, JHH5, and JHH7 cells) in cell culture system. In animal experiment, how the gene expression of Hip/Pap was affected by IH should be included.
Response: We thank Referee #3 for this suggestion. We verified RNAseq data with focus on HIP/PAP and Regenerating Family Member genes. Unfortunately, we did not observe any modulations, which I probably related to the type of experimental model and our selected protocol. As we now explain in the discussion, the tissues and blood samples were collected 12h after the last hypoxia cycle. Therefore we could not precisely evaluate the direct and transient effects of intermittent hypoxia. Reference Uchiyama, T. et al. (Biochem. Biophys. Rep. 11, 130-137, 2017) is now included in the discussion.
4 In Citation (lines 28-30), Autor(s)’ name and title of manuscript should be added.
Response: We apologize. The citation was corrected.
5 “IH” should be explained: intermittent hypoxia (IH). (line 65)
Response: We apologize for the unclear description. “Intermittent hypoxia” is now used in whole text and IH is used only as a description of group treated by intermittent hypoxia.
6 Although the authors showed approval number, they did not show approval date (lines 79-82). Please add the approval date.
Response: We thank Referee #3 for this comment. The exact dates have been included in the revised manuscript.
7 “ImageJ software (NIH, USA)” (line 135) should be changed to “ImageJ software (NIH, Bethesda, MD)”.
Response: We thank Referee #3 for this comment. We have adjusted this in the revised manuscript.
8 “RNAeasy” (line 139) should be changed to “RNeasy”
Response: We thank Referee #3 for this comment. We have adjusted this in the revised manuscript.
9 “Retrotranscription” (line 142) should be changed to “Reversetranscription”.
Response: We thank Referee #3 for this comment. We have adjusted this in the revised manuscript.
10 “Thermofisher” (lines 144, 145) should be changed to “Thermo Fisher”.
Response: We thank Referee #3 for this comment. We have adjusted this in the revised manuscript.
11 “GraphPad Software Inc., CA, USA” should be changed to “GraphPad Software Inc., San Diego, CA)”.
Response: We thank Referee #3 for this comment. The description of GraphPad Software was corrected.
12 There was no DOI in Ref. 15.
Response: We thank Referee #3 for this comment. The DOI was added.
Round 2
Reviewer 1 Report
The authors addressed all my concerns. Please check the manuscript carefully for typos and language issues before final publication.
Reviewer 3 Report
Judged by the revised version from the authors, all the points are suitably revised in the REVISED VERSION except for a minor point.
<Minor point>
1 “RNAeasy” (line 152) should be changed to “RNeasy”. Please see “https://www.qiagen.com/ja-us/products/discovery-and-translational-research/dna-rna-purification/rna-purification/total-rna/rneasy-plus-kits/”.